# Beyond Pattern Recognition: TLR2 Promotes Chemotaxis, Cell Adhesion, and Migration in THP-1 Cells

**DOI:** 10.3390/cells12101425

**Published:** 2023-05-19

**Authors:** Katrin Colleselli, Marie Ebeyer-Masotta, Benjamin Neuditschko, Anna Stierschneider, Christopher Pollhammer, Mia Potocnjak, Harald Hundsberger, Franz Herzog, Christoph Wiesner

**Affiliations:** 1Department of Medical and Pharmaceutical Biotechnology, IMC University of Applied Sciences, 3500 Krems, Austria; 2Department for Biomedical Research, University for Continuing Education Krems, 3500 Krems, Austria; 3Institute Krems Bioanalytics, IMC University of Applied Sciences, 3500 Krems, Austria; 4Gene Center Munich, Department of Biochemistry, Ludwig-Maximilians-Universität München, 80539 Munich, Germany

**Keywords:** Toll-like receptor 2, chemotaxis, cell adhesion, cell migration, THP-1

## Abstract

The interaction between monocytes and endothelial cells in inflammation is central to chemoattraction, adhesion, and transendothelial migration. Key players, such as selectins and their ligands, integrins, and other adhesion molecules, and their functions in these processes are well studied. Toll-like receptor 2 (TLR2), expressed in monocytes, is critical for sensing invading pathogens and initiating a rapid and effective immune response. However, the extended role of TLR2 in monocyte adhesion and migration has only been partially elucidated. To address this question, we performed several functional cell-based assays using monocyte-like wild type (WT), TLR2 knock-out (KO), and TLR2 knock-in (KI) THP-1 cells. We found that TLR2 promotes the faster and stronger adhesion of monocytes to the endothelium and a more intense endothelial barrier disruption after endothelial activation. In addition, we performed quantitative mass spectrometry, STRING protein analysis, and RT-qPCR, which not only revealed the association of TLR2 with specific integrins but also uncovered novel proteins affected by TLR2. In conclusion, we show that unstimulated TLR2 influences cell adhesion, endothelial barrier disruption, migration, and actin polymerization.

## 1. Introduction

Monocytes, which comprise approximately 10% of total human leukocytes, play a fundamental role in protective immunity and are involved in both the initiation and resolution of inflammation [1,2]. Early inflammation requires the recruitment of circulating blood monocytes across the endothelium to the site of infection or injury [3]. This multistep event begins with the activated endothelium expressing chemoattractants, cell adhesion molecules, and the molecules’ receptors, which attract monocytes [4]. As a result, the monocytes start to attach to the endothelium and loosely roll along the vascular surface. This is followed by their firm adhesion to the endothelium, which is also known as the arrest phase, and transendothelial migration (diapedesis). In the latter, the endothelial barrier is disrupted, and monocyte migration across the barrier is possible [5]. There are specific molecules that orchestrate this homing process. For endothelial cells, intercellular adhesion molecule-1 (ICAM-1) and vascular cell adhesion molecule-1 (VCAM-1) are prominent examples, whereas for monocytes, selectins such as L-selectin and integrins such as β2-integrin (ITGB2) or α4-integrin (ITGA4) are highly expressed for this process [6,7,8]. Adhesion involving integrins is tightly controlled by ligand binding, which allows rapid switching from the inactive state (low affinity binding) to the active state (high affinity confirmation), which is also known as inside-out signaling [9,10,11]. The active state allows integrins to bind ICAM-1 more efficiently, which is initially triggered by the stimulation of chemoattractant receptors or Toll-like receptors (TLRs) [12,13].

TLRs, which are among the best-known pattern recognition receptors, are key elements in fighting invading pathogens [14,15]. They are widely distributed in immune cells, such as dendritic cells or monocytes, and in other somatic cells, such as fibroblasts or endothelial cells [16]. Ten receptors (TLR1-10) have been identified in humans [17]. TLR2, together with TLR1, TLR4, TLR5, and TLR6, are expressed on the cell surface to recognize extracellular pathogen-associated molecular patterns (PAMPs) such as bacterial lipopolysaccharide (LPS). In contrast, TLR3, TLR7, and TLR9 are located intracellularly in endosomal compartments to spot microbial RNA or DNA [18]. The formation of heterodimers (TLR2/TLR1 or TLR2/TLR6) or homodimers (TLR4) via ligand binding activates transcription factors, such as nuclear factor-κB (NF-κB), which are released to translocate to the nucleus and induce the expression of various genes, including those responsible for inflammation and adhesion mediation [19].

Recent evidence suggests that TLRs, especially TLR2 and TLR4, are involved in integrin-dependent leukocyte adhesion and transmigration [13,20]. The latter is often implicated in atherosclerosis and atherosclerotic plaques, where monocyte adhesion becomes pathological [21,22,23,24]. One hypothesis is that the stimulation of monocytes by peptidoglycan (PG), which activates TLR2, affects the adhesion and migration properties of these cells [25]. Additionally, there is evidence that TLR2 activated by Listeria monocytogenes was a promoter of monocyte and macrophage migration and mobility [26].

All previous findings have indicated that TLR2, as an activated receptor, promotes the process of homing. Our study aims to show that there is a significant difference in cell adhesion and migration induced by TLR2, even in its inactive state, affecting the gene expression of specific proteins essential for these processes. Therefore, we stably transfected THP-1 TLR2 knock-out (KO) cells with human full-length TLR2 and compared THP-1 wild-type cells (WT), KO cells, and the generated TLR2 knock-in (KI) cells using various cell-based assays. We also performed proteomics and gene expression analysis with a focus on adhesion- and migration-associated proteins and examined the actin polymerization. We found that TLR2 per se, without prior activation, plays a significant role in the homing process, including chemoattraction, cell adhesion, and cell migration.

## 2. Materials and Methods

### 2.1. Cell Culture

The human acute monocytic leukemia-derived THP-1 cell line (WT) (ATCC^®^, Manassas, VD, USA), the THP-1-Dual™ KO-TLR2 cell line (KO) (InvivoGen, Toulouse, France), and the THP-1-Dual™ KI-TLR2 cell line (KI) (see Section 2.2) were maintained in RPMI-1640 (Thermo Fisher Scientific, Vienna, Austria) and supplemented with 2 mM L-glutamine, 10% heat-inactivated fetal calf serum (FCS), and 100 U/mL penicillin/streptomycin (everything Thermo Fisher Scientific). 293Ta (GeneCopoeiaTM, Rockville, MD, USA) were cultured in DMEM growth medium supplemented with 2 mM L-glutamine, 10% heat-inactivated FCS, and 100 U/mL penicillin/streptomycin. Human lymphatic endothelial cells (LECs), immortalized by ectopic expression of telomerase reverse transcriptase [27], were cultured in huMEC medium (InSCREENex, Braunschweig, Germany) supplemented with 100 µg/mL Normocin (InvivoGen). Human umbilical vein endothelial cells (HUVECs) (provided by the University for Continuing Education Krems, Austria) were isolated from umbilical cords of healthy donors as previously described [28]. HUVECs were cultured in M199 (Sigma-Aldrich, Saint-Louis, MO, USA), 20% FCS (Thermo Fisher Scientific), 10 µg/mL endothelial cell growth supplements (Merck, Kenilworth, NJ, USA), and 15 IU/mL heparin (Gilvasan, Vienna, Austria). Conditioned medium from LPS-stimulated whole blood was obtained by collecting blood from healthy donors after written consent. Blood was activated for 4 h at 37 °C with lipopolysaccharide (LPS) from *Escherichia coli* purified by gel-filtration chromatography (Sigma-Aldrich) at a final concentration of 100 ng/mL. After activation, blood was centrifuged at 2500× *g* for 15 min to isolate the plasma. Plasma was further diluted 10-fold in fresh medium (M199 or huMEC) to obtain conditioned medium.

### 2.2. Cloning and Lentiviral Transduction

To generate a TLR2 knock-in cell line (KI), full-length human TLR2 was cloned into the lentiviral vector pEZ-Lv195 with a plasmid size of 9784 base pairs and a puromycin selection marker (GeneCopoeiaTM). The construct was then transformed into the *Escherichia coli* strain GCI-5α (GeneCopoeiaTM) and verified by sequencing. DNA sequencing was carried out by Microsynth Austria GmbH with the following primers: primer sequence 1: GTTTCGTTTTCTGTTCTGC; sequence 2: CGGAGGCTGCATATTCCAAG; sequence 3: TGTGTCTTCATAAGCGGGACT.

For lentiviral transduction, 293Ta lentiviral packaging cells and the Lenti-Pac™ HIV Expression Packaging Kit (both GeneCopoeiaTM) were used according to the manufacturer’s instructions. Supernatant containing viral particles was used for infection or stored at −80 °C. For transfection, 1 × 10^6^ THP-1 KO cells were infected with 1 mL of virus suspension that was diluted in complete culture medium with 8 µg/mL polybrene. After 24 h, the medium was replaced with complete cell culture medium without polybrene, and 48 h later, selection was started with 1 µg/mL puromycin dihydrochloride (Thermo Fisher Scientific).

### 2.3. Western Blotting

For the preparation of whole cell lysates, a total number of 2 × 10^6^ cells were harvested for each cell line (KO, WT, and KI). Cells were then washed with PBS and lysed in 100 µL ice-cold lysis buffer consisting of 500 mM NaCl, 50 mM Tris-HCl, pH 7.4, 0.1% SDS, 1% NP-40 and 0.05% NaN3, 1U DNase I, 1 U protease, and phosphatase inhibitor cocktail (all from Thermo Fisher Scientific). Cell lysates were frozen at −80 °C for 2 h and centrifuged at 12,000 rpm for 30 min at 4 °C, and the supernatants were resuspended in 4X Laemmli sample buffer (Bio-Rad, Hercules, CA, USA) containing 10% ß-mercaptoethanol (Merck). Proteins were separated on 7.5% Mini-PROTEAN TGX Precast Protein Gels (Bio-Rad) using PageRuler™ Plus Prestained Protein Ladder (Thermo Fisher Scientific) as a molecular weight marker and run at 100 V. The gels were then transferred onto 0.2 µm nitrocellulose membranes (Bio-Rad) using a Trans-Blot^®^ Turbo Transfer System (Bio-Rad). To saturate non-specific binding sites, immunoblots were blocked with 5% nonfat dry milk (New England Biolabs, Frankfurt, Germany) in PBS supplemented with 0.1% Tween 20 (PBS-T) overnight at 4 °C. Primary anti-TLR2 antibody (1:1000; Invitrogen, Waltham, MA, USA) and vinculin (1:200; Santa Cruz Biotechnologies, Dallas, TX, USA) were diluted in 5% nonfat dry milk in PBS-T, and the membranes were incubated for 2 h at room temperature. After thorough washing with PBS-T, the blots were incubated for 1 h with secondary HRP-conjugated anti-rabbit antibody (Thermo Fisher Scientific). Membranes were washed with PBS-T and developed with the Clarity Western ECL Substrate (Bio-Rad) according to the manufacturer’s instructions. Protein visualization was performed using ChemiDoc MP Imaging Systems with Image Lab^TM^ (Bio-Rad).

### 2.4. Adhesion Assays

LECs were seeded at a density of 0.25 × 10^6^ cells/mL in a 96-well plate, and, after reaching 90% confluence, they were stimulated with conditioned media from LPS-stimulated whole blood for 4 h. An amount of 0.5 × 10^6^ cells/mL of THP-1 KO, WT, or KI cells was labeled with Hoechst 33342 (2 µg/mL, Thermo Fischer Scientific) and added to the LECs for 15 min under shaking conditions. After thorough washing, fluorescence was measured using the Mini Max 300 imaging cytometer (Molecular Devices, LLC, San Jose, CA, USA).

To measure adhesion under flow, HUVECs were seeded onto fibronectin-coated channel slides (ibidi, Gräfelfing, Germany) at a density of 1 × 10^6^ cells/mL and cultured as previously described [29]. Briefly, cells were allowed to adhere under static conditions for 2 h at 37 °C in humidified atmosphere (5% CO_2_), and slides were then connected to fluidic units and cultured for 1 h at 2 dyn/cm^2^. The flow rate was further increased to 5 dyn/cm^2^. After reaching 100% confluence, the cells were activated with conditioned medium for 4 h at a shear stress of 5 dyn/cm^2^. After endothelial activation, the reservoirs were spiked with either THP-1 WT, KO, or KI cells to reach a final concentration of 0.5 × 10^6^ cells/mL in the reservoirs. Monocyte adhesion was assessed using the CKX-41 inverted cell culture microscope and cellSens software (both Olympus, Tokyo, Japan), with images taken every 15 s for 15 min.

### 2.5. Chemotaxis Assay

To evaluate and compare the migration capacity of THP-1 WT and KO cells, Transwell^®^ 24-well permeable plates with 5 µm pore polycarbonate membranes (Szabo-Scandic, Vienna, Austria) were used for the migration assay. First, LECs were seeded into T25 cell culture flasks (Sarstedt, Nümbrecht, Germany), and, after reaching 90% confluence, the cells were washed with PBS. The cells were incubated in huMEC media without FCS and treated with or without LPS (100 ng/mL) for 4 h before the media was applied to the lower compartment of the transwell chambers. THP-1 cell lines (KO and WT) with a density of 1 × 10^6^ cells/mL were stained with Hoechst 33342 (2 µg/mL), added to the upper chamber of the transwell plate, and incubated for 2 h to allow the cells to migrate through the filter. After aspiration of the remaining cells, the cells adhering to the filter were gently removed with a cotton swab, and the migrated cells (bottom of the filter) were visualized with an inverted microscope (DMI6000 B, Leica, Mannheim, Germany) using a 40× objective and analyzed with the Leica Application Suite Version X 3.8.0 software.

### 2.6. Transmigration Assay

Endothelial cell monolayer disruption (transmigration) by monocytes +/− LPS and TLR2 ligand stimulation was measured using the 9600Z Electrical Cell-Substrate Impedance Sensing (ECIS) system (Applied BioPhysics, Troy, NY, USA). A 96-well plate containing 20 gold film electrodes per well (ibidi) was coated with 1 mg/mL neutralized rat tail collagen type I (Thermo Fisher Scientific) for 10 min at room temperature and 2 µg/mL bovine plasma fibronectin (Thermo Fisher Scientific) for 45 min at 37 °C and 5% CO_2_. Wells were then inoculated with LECs at a seeding density of 30,000 cells per well in 100 µL huMEC complete media at 37 °C and 5% CO_2_. The run was performed under a single frequency of 4000 Hz and continuously monitored every 60 s. Impedance (Z) was determined by Ohm’s law: Z = V/I. After 20 h, when an endothelial monolayer was reached, KO and WT THP-1 cells with and without stimulants were added at a density of 100,000 cells per well in 50 µL huMEC complete media. For stimulation, 100 ng/mL LPS (Thermo Fisher Scientific) or 300 ng/mL of either TLR2-specific ligand Pam2CSK4 (Pam2) or Pam3CSK4 (Pam3) (both InvivoGen) was used. Endothelial monolayer disruption was observed for 10 h.

### 2.7. Proteomics

For quantitative mass spectrometry, THP-1 KO, WT, and KI cells were seeded at a density of 1 × 10^6^ cells per well in a 12-well plate and incubated for 3 h at 37 °C and 5% CO_2_. The cells were then collected and washed with PBS twice.

For sample preparation and digestion, the cell pellets were dissolved in 40 µL lysis buffer containing 8 M urea and 50 mM NH₄HCO₃ (both Sigma-Aldrich) and sonicated for 10 min. After centrifugation at 14,000× *g* for 10 min, the supernatants were transferred to new tubes to remove most of the DNA and stored at −20 °C until further use.

After protein concentration was determined using the BCA assay (Sigma-Aldrich), 20 µg of protein per sample was used for digestion. Samples were reduced and alkylated with TCEP and IAA and sequentially digested with Lys-C (FUJIFILM Wako Chemicals U.S.A. Corporation, Richmond, VA, USA) for 1 h and trypsin (Promega, Walldorf, Germany) for 16 h. Peptides were purified using Sep-Pak tC18 1 cc Vac cartridges (Waters, Vienna, Austria), dried, and stored at −20 °C until analysis.

Samples were analyzed using an Ultimate 3000 RSLCnano system coupled with an Orbitrap Eclipse Tribrid mass spectrometer (both Thermo Fisher Scientific).

Dried samples were suspended in 40 µL of mobile phase A (98% H_2_O, 2% ACN, and 0.1% FA). A total of 2 µL was injected into a PepMap 100 (C18 0.3 × 5 mm) TRAP column and analyzed using a PepMap RSLC EASY-spray column (C18, 2 µm, 100 Å, 75 µm × 50 cm, Thermo Fisher Scientific). Separation was performed at 300 nL·min^−1^ with a flow gradient of 2–35% mobile phase B (2% H_2_O, 98% ACN, and 0.1% FA) within 60 min, resulting in a total method time of 80 min. The mass spectrometer was operated in DIA mode with the FAIMS Pro system in positive ionization mode at CV-45. MS1 scans were acquired in the scan range of 350–1400 *m*·*z*^−1^ with a resolution of 120,000 @200 *m*·*z*^−1^. For DIA scans, the precursor mass range was set to 400–1000 *m*·*z*^−1^ with a 14 *m*·*z*^−1^ isolation window and 1 *m*·*z*^−1^ window overlap for a total of 43 independent scans. HCD fragmentation was performed at 30% NCE, and fragments were analyzed in the Orbitrap at a resolution of 30,000 @200 *m*·*z*^−1^.

To deepen the analysis, a pool of all samples was created and used for gas phase fractionation (GPF) [30]. Sample pool was analyzed 6 times consecutively with smaller precursor mass ranges of 100 *m*·*z*^−1^ (400–500, 500–600, 600–700, 700–800, 800–900, and 900–1000 *m*·*z*^−1^) and isolation windows of 4 *m*·*z*^−1^ with 2 *m*·*z*^−1^ window overlap.

DIA-NN (version 18.1.1) [31] was used for protein identification and quantification. The GPF samples were first searched against the human protein database (Uniprot, version 10.2021, 20,386 entries), and a spectral library was created using the identified peptides. The main samples were searched using the spectral library together with the human FASTA database to maximize the number of identified proteins. Perseus (version 2.0.6.0) [32] was used for statistical evaluation. Protein groups were filtered according to their treatment, requiring at least three out of three values to be valid in at least one group. Remaining missing values were replaced with a downward shift of 1.8 σ and a width of 0.3 to allow statistical testing for all remaining protein groups.

Furthermore, upregulated genes in THP-1 WT and/or KI vs. KO cells associated with cell adhesion + TLR2 were analyzed for interactions using the © STRING CONSORTIUM 2023 (version 11.5). This generated protein–protein network shows interactions based on data from experiments, genomic context predictions, automated text mining, co-expressions, and curated databases [33].

The mass spectrometry proteomics data have been deposited to the ProteomeXchange Consortium via the PRIDE [34] partner repository with the dataset identifier PXD041819.

### 2.8. Real-Time Quantitative PCR (RT-qPCR)

Total RNA was isolated using the RNeasy kit (Qiagen, Vienna, Austria), and 1 µg RNA was reverse transcribed using the Hight Capacity cDNA Reverse Transcription kit (Thermo Fisher Scientific), both according to the manufacturer’s instructions. Pre-designed TaqMan gene expression assays with FAM-labeled primers/probes (Applied Biosystems, Foster City, CA, USA) and TaqMan^®^ Gene Expression Master Mix (Thermo Fisher Scientific) were used for quantification. Amplification parameters were 10 min at 95 °C for initial denaturation, followed by 40 cycles of 95 °C for 20 s and 60 °C for 1 min run on the Quant Studio 7 Flex (Applied Biosystems). The fold change in the mRNA expression was determined using the ΔΔCt method [35].

### 2.9. Confocal Immunofluorescence Microscopy

F-actin expression was visualized by confocal microscopy. Briefly, 8-chamber glass slides (ibidi) were precoated with 11 µg/mL fibronectin (Thermo Fisher Scientific). THP-1 WT, KO, and KI suspensions (4 × 10^5^ cells/mL) were added to each well, and cells were allowed to attach overnight at 37 °C and 5% CO_2_. The cells were washed with PBS and fixed for 10 min with a 4% paraformaldehyde solution at room temperature. The acidity of the paraformaldehyde was neutralized with a 50 mM ammonium chloride solution (Sigma-Aldrich) for 10 min at room temperature. Cells were washed with PBS and permeabilized with 0.5% Triton X-100 (Sigma-Aldrich) for 10 min at 4 °C and blocked with 5% goat serum (Invitrogen) for 30 min at room temperature. Cells were washed with PBS, and F-actin was stained with phalloidin conjugated to AlexaFluor 635 (Invitrogen) diluted 1:200 in 1% bovine serum albumin solution (Sigma-Aldrich) for 60 min at room temperature in the dark. Cells were counterstained with DAPI (Sigma-Aldrich) and mounted with Fluoromount aqueous mounting medium (Sigma-Aldrich) onto high-precision microscope coverslips (1.5H, Marienfeld, Lauda-Königshofen, Germany). Images of the stained cells were acquired with a confocal laser scanning microscope (TCS SP8, Leica) using a 63X glycerol objective (numerical aperture 1.3). The 3D images of the beads were obtained by z-stack imaging (acquisition of 129 Z-stack steps; step size: 0.33 µm; scan speed: 700Hz; resolution: 1024 × 1024). Image analysis was performed using the LAS X software (Leica).

### 2.10. Flow Cytometric Analysis

THP-1 WT, KO, and KI cell suspensions were harvested (1 × 10^6^ cells/mL), washed, fixed, and permeabilized as described above (Section 2.9). F-actin polymerization was measured by applying AlexaFluor 635-conjugated phalloidin (Thermo Fisher Scientific) diluted 1:200 in 1% bovine serum albumin-PBS for 60 min at room temperature and analyzed using the BD Accuri™ C6 Plus flow cytometer and BD Accuri C6 Plus software (BD Life Sciences, San Jose, CA, USA).

### 2.11. Statistics

Statistical analyses and graphs were generated using GraphPad Prism (version 9.03, San Diego, CA, USA), and the proteomics data were evaluated using Perseus (version 2.0.6.0). Data are expressed as mean ± standard deviation and significance was accepted at *p* ≤ 0.05. One- or two-way ANOVA with post hoc Dunnett’s multiple comparison test were used to assess differences between multiple groups, and Student’s t-test was performed to compare data between two groups.

## 3. Results

### 3.1. TLR2 Enhances Chemotactic Migration, Adhesion, Endothelial Barrier Disruption, and Transendothelial Migration

To investigate whether TLR2 plays an important part in rolling, adhesion, and cell migration/transmigration during inflammatory events, we used a monocytic THP-1 WT, a KO, and a stable KI cell line. As indicated in Figure 1a, the THP-1 KI cells show a clear but much lower TLR2 protein expression than the WT cells. However, no TLR2 expression could be detected in the KO cells (Figure 1a). Since previous studies have shown that TLR2 activation can promote leukocyte migration, adhesion, and transmigration in vitro, we wanted to determine whether unstimulated TLR2 per se plays a role in this homing process [13,20,26,36,37]. Because the LPS treatment of endothelial cells induces the expression of various inflammatory mediators including interleukin-6 (IL-6), IL-8, and ICAM-1 [38,39], we first stimulated LECs with LPS (100 ng/mL) for 4 h and used the supernatant as a chemoattractant for THP-1 cell migration. The THP-1 WT and KO cells were stained with Hoechst 33342 and seeded in the upper chamber of the 24-well permeable transwell plates and allowed to migrate for 2 h through the 5 μm pore size filters towards the supernatants of LPS-treated or untreated endothelial cells in the lower chambers of the transwells. As shown in Figure 1b,c, the WT cells exhibited significantly higher chemoattraction toward activated endothelial cell supernatants than the KO cells. Next, we performed an adhesion assay in which a monolayer of endothelial cells (LECs) in a 96-well plate was stimulated with LPS for 4 h, and Hoechst-stained THP-1 cells were cocultured with endothelial cells under shaking conditions for 15 min. After medium aspiration and thorough washing, the fluorescence levels of those cells that had formed a tight attachment to the endothelial cells were measured. As expected, we found a significant increase in THP-1 WT adhesion and a modest increase in THP-1 KI adhesion in activated endothelial cells compared to KO cells. (Figure 1d). To confirm these results under more physiological conditions, we then performed an adhesion assay under flow with primary HUVECs. First, cells were treated with a conditioned medium from LPS-stimulated whole blood for 4 h, and the activation of the HUVECs was confirmed by measuring the increase in ICAM-1 expression (Figure 1e). After endothelial activation under flow, the THP-1 WT, KO, and KI cells were added to the reservoirs, and adhesion was recorded for 15 min by brightfield microscopy. As clearly shown in Appendix A and Figure 1f, THP-1 WT and KI cells adhere faster and more firmly to the endothelial cell layer, whereas KO cells appear to roll longer on the layer and/or detach more easily and exhibit weaker adhesion. In contrast to the LPS-stimulated HUVECs, no adhesion could be detected in the non-activated cells (Appendix A). Thus, we have clearly demonstrated here that TLR2 alone, even without prior stimulation with a specific TLR2 ligand, plays an essential role in migration and adhesion to activated endothelial cells in THP-1 cells.

To investigate whether TLR2 also induces endothelial barrier disruption, and the transendothelial migration of THP-1 WT and KO cells, endothelial cells were plated on a 96-well plate with gold film electrodes (ECIS), precoated with collagen type I and fibronectin, and allowed to form a monolayer. Cells were left untreated or were treated with LPS, which is a potent stimulator of TLR4, Pam2, and Pam3, which are TLR2 ligands, and THP-1 KO and WT cells were added directly to the endothelial cells. Endothelial monolayer resistance was then measured by real-time impedance quantification, demonstrating changes in endothelial barrier disruption. As shown in Figure 2a–h, endothelial monolayer disruption was only observed when cells were stimulated with Pam2, Pam3, or LPS. Interestingly, THP-1 WT cells significantly enhanced endothelial monolayer disruption compared to KO cells (Figure 2a–h). Appendix A demonstrates that the endothelial barrier disruption with THP-1 KI cells is not as intense as with WT cells but is stronger than with KO cells, which is explained by the lower TLR2 expression. These results clearly demonstrate that TLR2 expressed on THP-1 cells has an important function in endothelial barrier disruption and transendothelial migration.

### 3.2. Proteomics Reveals Novel TLR2-Dependant Proteins Involved in Adhesion and Migration

To gain more insight into the protein expression affected by TLR2, we performed label-free quantification mass spectrometry. For this purpose, THP-1 WT, KO, and KI cells were harvested, and a data-independent acquisition (DIA) method was applied to compare THP-1 WT and KI vs. KO cells. In general, approximately 7200 proteins were identified in our samples, of which 858 proteins were significantly regulated when comparing WT vs. KO cells, and 951 proteins were significantly regulated when comparing KI vs. KO cells (FDR = 0.05, s0 = 1). Furthermore, 236 common proteins were found to be significantly upregulated in WT and KI vs. KO cells. (Appendix A). Relevant proteins that appeared to be upregulated in WT and KI cells were highlighted (Figure 3a,b). TLR2 was significantly upregulated in both THP-1 WT and KI cells compared to KO cells, but the expression was much stronger in WT cells, which was consistent with our immunoblot result in Figure 1a. Certain proteins such as integrin β1 (ITGB1), integrin β2 (ITGB2), and integrin αL (ITGAL) were found to be significantly upregulated only in WT and not in KI cells (Figure 3d,f,g). Among the integrins, only integrin αX (ITGAX) was upregulated in both WT and KI compared to KO cells (Figure 3e). Other molecules that play an essential role in adhesion, such as galectin-3 (LGALS3), galectin-3-binding protein (LGALS3BP), and vitronectin (VTN), were highly upregulated in WT and KI cells in comparison to KO cells (Figure 3h–j). CD44 was significantly upregulated in WT vs. KO but not at all in KI vs. KO cells (Figure 3k). Transforming growth factor-beta-induced protein ig-h3 (TGFBI) and myosin 1G (MYO1G) were highly expressed in both WT and KI cells (Figure 3l,m). Protein disulfide-isomerase beta-subunit of prolyl 4-hydroxylase (P4HB), which is involved in cell migration [40], was found to be upregulated in KI and WT cells (Figure 3n). In summary, we showed that certain integrins are dependent on TLR2 expression and identified novel proteins such as MYO1G and P4HB that are influenced by TLR2.

### 3.3. Adhesion Molecule Interaction Network of TLR2 Mapped by STRING Analysis

To obtain a comprehensive and objective connectivity network of TLR2, including both direct and indirect interactions, a STRING network analysis was performed. For this purpose, the highly expressed proteins in THP-1 WT and/or KI vs. KO cells related to cell adhesion and migration were used. This network determines interactions based on data from experiments, genomic context predictions, automated text mining, co-expressions, and curated databases. As shown in Figure 4, we found 10 nodes (representing proteins) and 20 edges (representing protein–protein interactions). Interestingly, strong protein–protein interactions were identified between TLR2 and the integrins ITGAL, ITGAX, ITGB2 and between TLR2 and LGALS3 and CD44.

### 3.4. Regulation of TLR2-Dependent mRNA Levels Does Not Always Match That at the Protein Level

To deepen our understanding of the influence of TLR2 on gene expression, we examined the mRNA expression levels of certain molecules that were upregulated in THP-1 WT and/or KI cells in the proteomics results shown above. The expression of the integrins ITGAL, ITGAX, and ITGB1 reflected a similar pattern to that seen in the proteomics data. However, the increase between THP-1 WT and/or KI cells and KO cells was less pronounced, often not significant, or not present at all (Figure 5a–c). A completely different expression was observed for ß2 integrin (Figure 5d). In contrast to the proteomics data, the mRNA level was significantly decreased in cells expressing TLR2. Furthermore, the mRNA level of CD44 was significantly increased not only in WT cells but also in the KI cells, and LGALS3 exhibited no difference between TLR2-expressing and KO cell lines (Figure 5e,f). However, a less pronounced correlation between transcriptomic and proteomic data is not unexpected and has been observed before, especially since post-translational modifications are not investigated in the analysis [41,42,43].

### 3.5. TLR2 Is Involved in Regulation of Actin Polymerization

Actin reorganization and polymerization are critical for leukocyte adhesion [44] and transendothelial migration [45]. Therefore, we investigated the role of TLR2 in actin polymerization in THP-1 cell lines. THP-1 KO, WT, and KI cells were harvested, fixed, permeabilized, and stained with AF635-conjugated phalloidin, which can selectively bind to F-actin. Cells were analyzed by confocal microscopy and FACS. In contrast to KO cells, WT and KI cells showed increased F-actin staining (Figure 6a–c). This was further confirmed by the FACS analysis, in which THP-1 WT cells exhibited more than a 30% increase in F-actin, and KI cells showed 10% more F-actin than KO cells (Figure 6d–f). In addition, our proteomics data revealed three proteins involved in actin regulation and organization in TLR2-expressing THP-1 cells: MICAL-like protein 2 (MICALL2), Ras-related protein R-Ras (RRAS), and tetratricopeptide repeat protein 17 (TTC17) (Figure 6a–c).

## 4. Discussion

The importance of TLR2, not only as a pattern recognition receptor but also in cell adhesion and migration with a focus on TLR2 ligand activation, has been clearly demonstrated [13,20,26]. This in vitro study was designed to investigate the fundamental role of TLR2 expressed in monocyte-like cells on activated endothelial cell monolayers in terms of chemotaxis, adhesion, endothelial barrier disruption, and transmigration using THP-1 wild type WT, KO, and KI cells. The mechanisms of chemotaxis and the regulation of chemotactic factors for monocyte migration, leading to tissue infiltration and monocyte differentiation in macrophages or dendritic cells, has already been extensively studied [46]. Our chemotaxis assay, focusing on the role of TLR2, showed significantly increased migration in TLR2-expressing THP-1 cells, in contrast to KO cells, when using the supernatant of LPS-stimulated endothelial cells. In addition to LPS-induced cytokine and chemokine production, residual LPS alone in the endothelial cell supernatant can activate pattern recognition receptors such as TLR4, TLR2, and the nucleotide binding oligomerization domain-containing 1 (NOD1) receptor expressed on THP-1 cells [47,48,49]. To this end, MyD88-mediated LPS/TLR4 crosstalk with TLR2 has also been described to enhance and stabilize NF-κB and ICAM-1 expression [50]. However, unlike primary monocytes, THP-1 cells express very low levels of CD14, and, consequently, THP-1 cells hardly respond to LPS via the TLR4 pathway [51]. Furthermore, we assume that the expression levels of the aforementioned receptors are the same in our THP-1 cell lines, except for TLR2, since no differences were detected in our proteomic analysis. This supports the theory that TLR2 is an essential chemoattractant sensor and consequently contributes to cell adhesion and migration [52]. A 2015 study showed that TLR2 increased the susceptibility of endothelial monolayers to disruption in an in vitro assay [53]. By using functional cell-based assays that measure endothelial barrier disruption and transendothelial migration, we were able to demonstrate a quicker and more pronounced endothelial monolayer breakage when using THP-1 WT cells compared to when using KO cells. Interestingly, significant differences between KO and WT cells were observed not only after Pam2 and Pam3 stimulation but also after LPS stimulation, suggesting that general endothelial activation and the addition of THP-1 cells are sufficient for endothelial cell monolayer disruption. However, TLR2 enhanced endothelial monolayer disruption independent of the ligand that promoted endothelial monolayer activation. Furthermore, we observed stronger adhesion to activated endothelial cells in TLR2-expressing THP-1 cells compared to KO cells. In particular, using the more physiologically relevant adhesion assay under flow, we showed that TLR2-expressing cells exhibited a stronger and also faster adhesion to the endothelium, whereas the KO cells showed prolonged rolling and weaker adhesion. Leukocyte rolling is promoted by selectins and their ligands, such as P-selectin ligand-1 (PSGL-1) and E-selectin ligand-1 (ESL-1), followed by chemokine interactions, integrin activation, and leukocyte arrest [54]. The selectin ligands PSGL-1 and ESL-1, as well as CD44, possess specialized adhesive properties during tethering and rolling [55]. PSGL-1 and CD44 induce signals that activate the lymphocyte function-associated antigen α1 (LFA-1), which consists of the αL chain and β2. ESL-1, on the other hand, induces signals that activate the macrophage-1 antigen (Mac-1), which is composed of the subunits integrin αM and β2 [55]. The altered protein expression levels of selectin ligands and integrins could explain the prolonged rolling and weaker adhesion of THP-1 KO cells.

Therefore, we next performed quantitative mass spectrometry to investigate the protein expression patterns affected by TLR2 in THP-1 cell lines. Compared to in KO cells, we found that ITGAL, ITGAX, ITGB1, and ITGB2 were significantly upregulated in WT cells. In KI cells, in which TLR2 was expressed at much lower levels than in WT cells, all of the above integrins showed increased expression, but only ITGAX was significantly upregulated. Both ITGAX and ITGAL belong to the β2 integrins and are upregulated in monocytes during inflammation, resulting in increased cell adhesion to the endothelium [56]. An in vitro study and a mouse model lacking ITGAX recently demonstrated that ITGAX participates in human monocyte arrest in endothelial cells by cooperating with very late antigen-4 (VLA-4) in binding to VCAM-1 [57]. However, we did not find significant regulations with selectin ligands (e.g., PSGL-1), which would confirm the hypothesis that TLR2 does not seem to play a role in the interaction between selectins, since we still observed the capture and rolling of the monocytes on the surface of the endothelial cells [58]. In their study, Chung et al. (2014) have clearly demonstrated that TLR2 and TLR5 ligation can rapidly lead to the conformation of high-affinity β2 integrins that promote leukocyte adhesion to ICAM-1 and fibronectin. Accordingly, TLR2 activation by Pam3 increased integrin-dependent slow rolling and adhesion to the endothelium [20]. Other results have suggested that TLR2 activation induces cell migration that is specifically mediated by β1-integrin-induced pathways [36].

Direct interactions between TLR2 and all integrins, except for ITGB1, were found in the STRING analysis. We observed the same pattern in the mRNA expression levels of the integrins ITGAL, ITGAX, and ITGB1 as in the proteomics data. However, the upregulation between THP-1 WT and/or KI cells and KO cells was less pronounced, often not significant, or absent, and a completely different expression was observed for ITGB2. In contrast to the proteomics data, the mRNA level was significantly decreased in cells expressing TLR2. This led us to hypothesize that the heterogeneity may be due to post-translational modifications and/or alternative splicing, which is not an uncommon event to observe [41,43]. More than 400 types of modifications have been identified in proteins involved in adhesion and migration, including integrins [59]. Overall, these results show that TLR2, even in the absence of ligand activation, affects the protein expression of certain integrins.

In addition to integrins, the proteomic analysis and the subsequent protein network analysis highlighted several significant proteins involved in cell adhesion and migration, including CD44, LGALS3, LGALS3BP, VTN, and MYO1G. It has been shown that monocyte rolling under inflammatory conditions is strongly dependent on CD44 expressed in monocytes [60,61]. CD44, a hyaluronan receptor, mediates cell–cell and cell–matrix interactions, which are also critical for cell migration [62,63]. We found that CD44 was significantly elevated in WT vs. KO but not at all in KI vs. KO cells, whereas the mRNA levels were highly increased in WT and KI cells, and protein–protein interactions were found in the STRING analysis. There is growing evidence that CD44 affects TLR2 signaling [64,65]. However, since we saw that CD44 is only upregulated in WT THP-1 and not in KI cells, this may suggest that a certain amount of TLR2 is required for higher CD44 protein expression. A recent study revealed that TLR2 engagement alters CD44 expression and modulates the alternative splicing of CD44 pre-mRNA, which may explain the poor correlation between mRNA and protein levels [66]. Lectins, including LGALS3, are known to mediate adhesive interactions during leukocyte homing and directly impact the migratory behavior of these cells [67]. Polli et al. (2013) have indicated that LGALS3 is a driver of monocyte migration via interactions with extracellular matrix proteins such as fibronectin [68]. Our proteomics data demonstrate that both LGALS3 and LGALS3BP are highly upregulated in THP-1 WT and KI cells. The STRING protein analysis revealed a direct interaction between LGALS3 and TLR2 and an indirect interaction between LGALS3BP and TLR2 via LGALS3. Interestingly, we did not find a significant upregulation of LGALS3 mRNA levels, leading us to conclude that the influence of TLR2 is from protein modifications and, consequently, stabilization. A highly significant upregulation of VTN was detected in TLR2-expressing THP-1 cells. The STRING analysis revealed an indirect interaction of VTN with TLR2 via ITGB1, ITGB2, and CD44. This is not surprising since VTN is a glycoprotein found on the extracellular matrix that promotes cell adhesion via various glycosaminoglycans and integrins, thereby enhancing cell adhesion and migration [69,70]. TGFBI, a secretory protein induced by transforming growth factor-beta, has been shown to increase cell adhesion and spreading in fibroblasts [71]. With our proteomics data, we see a strong increase in TGFBI in the WT and KI cells compared to the KO cells, which suggests an effect of TLR2. Cruz-Zárate et al. (2021) recently showed that the absence of MYO1G results in defects in adhesion and chemotaxis [72]. MYO1G is a class I myosin found in many immune cells. It can interact with both actin filaments and the plasma membrane, thereby regulating cell motility, cell shape, and other cellular properties [73]. Our proteomic analysis revealed a strong upregulation of MYO1G in WT and KI THP-1 cells compared to KO cells. In addition, the STRING protein analysis indicated an indirect interaction with TLR2 via integrin LFA-1. The main function of the membrane protein P4HB is to catalyze the formation of disulfide bonds [74]. It has been suggested that P4HB is critical for cell adhesion and cell migration because it can modify the structure of proteins both inside and outside the cell [75]. Via its interaction with the receptor for galectin-9, P4HB remains on the cell surface of Th2 T helper cells, which enhances disulfide reductase activity at the plasma membrane, modifies the redox state of the plasma membrane, and promotes cell migration [40]. By showing a strong increase in P4HB in TLR2-expressing THP-1 WT and KI vs. KO cells, our proteomics results provide a novel link between TLR2 and P4HB in the context of cell adhesion and migration.

The coordinated polymerization of F-actin plays a fundamental role in proper whole cell migration [76]. Wang et al. (2019) demonstrated that TLR2 stimulation resulted in cellular motility and F-actin polymerization [26]. However, the mechanisms and proteins involved in the regulation of F-actin polymerization by TLR2 are still unclear. By comparing the three THP-1 cell lines, we found clear differences in F-actin expression levels. THP-1 WT cells displayed 30% more F-actin and KI cells (which expressed less TLR2) displayed 10% more F-actin compared to KO cells. We identified three proteins that were upregulated in WT and/or KI cells, which could be responsible for the alteration in F-actin. Highly upregulated MICALL2, a Rab13 effector protein, binds directly to F-actin where it can accelerate F-actin bundling and stabilization [77,78]. The small GTPase RRAS is also upregulated in TLR2-expressing cells, but the upregulation is not statistically significant (*p*-value 0.06). RRAS was first recognized because an overexpressed mutant of RRAS made cells highly integrin-dependent and adherent [79]. Furthermore, it has been indicated that RRAS interacts locally with the actin cytoskeleton, the actin-binding protein filamin A, to influence cell adhesion, cytoskeletal reorganization, and other processes that need navigation, such as leukocyte chemotaxis and migration [80,81]. A relatively uncharacterized protein is TTC17, which is located in the actin cytoskeleton, cytosol, and plasma membrane and participates in actin filament polymerization and cilium organization [82]. The identification of these three proteins in a TLR2-dependent manner supports our hypothesis that TLR2 expression influences actin polymerization, which then leads to the promotion of cell adhesion and mobility and transendothelial migration.

Overall, by comparing these three monocytic cell lines, we were able to demonstrate that TLR2 is a key mediator of cell adhesion and cell migration to the endothelium, even without prior stimulation. We discovered altered mRNA and protein expression patterns in several adhesion molecules, such as integrins, including ITGB1 and ITGAL, CD44, and LGALS3, that are dependent on TLR2 expression. In addition, novel TLR2-dependent proteins, such as P4HB and MYO1G, which are essential for the homing process, were uncovered, and new connections between TLR2 and actin polymerization were identified. Our results suggest that TLR2 in monocytes may be an important contributor not only in the initiation of inflammation but also in adhesion and cell migration by influencing the protein expression of integrins, adhesion molecules, and actin regulatory proteins.

## Figures and Tables

**Figure 1 cells-12-01425-f001:**
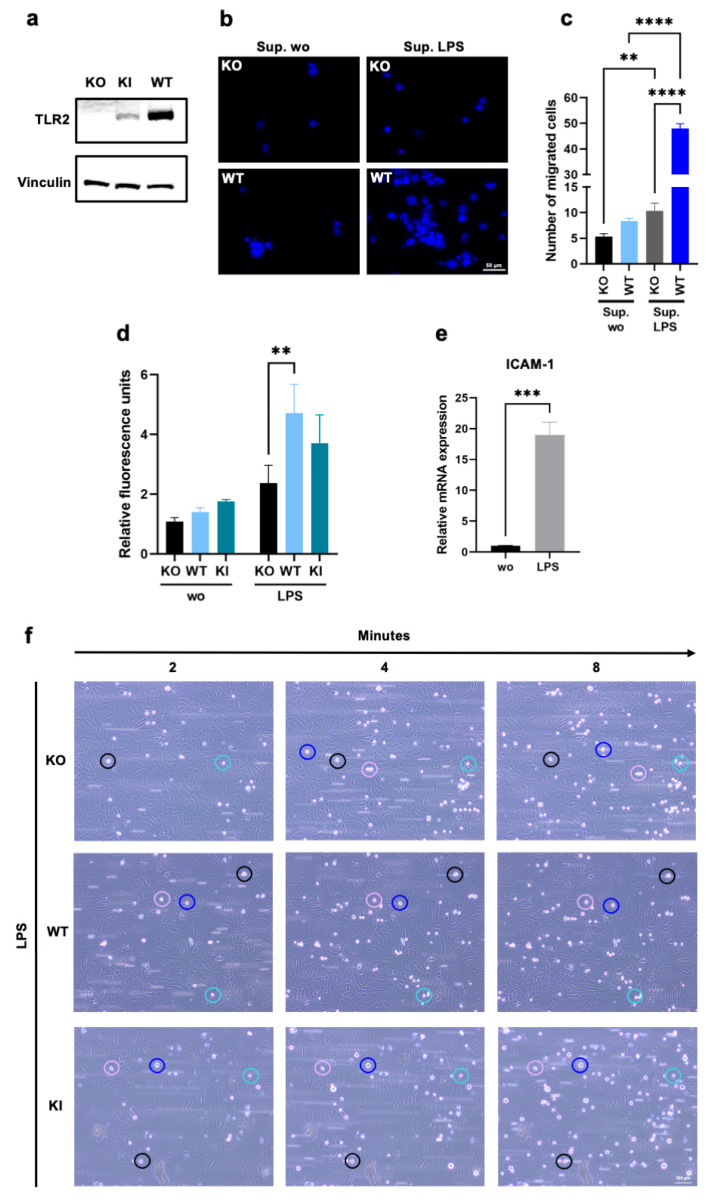
Chemotactic migration and adhesion of THP-1 wild type (WT), TLR2 knock-out (KO), and TLR2 knock-in (KI) cells. (**a**) Western blot analysis using an antibody against Toll-like receptor 2 (TLR2) in THP-1 KO, WT, and KI whole cell lysates. Vinculin was used as a loading control. (**b**,**c**) Cells were stained with Hoechst 33342, seeded in the upper chambers of 24-well transwell plates, and allowed to migrate through a 5 µm porous membrane for 2 h towards the supernatants of lipopolysaccharide (LPS)-treated or untreated human lymphatic endothelial cells (LECs). (**b**) Migration of THP-1 KO and WT cells was visualized by fluorescence microscopy. The scale bar represents 50 µm, and Hoechst 33342 is shown in blue. (**c**) Bar graphs show mean values ± standard deviation (n = 5). One-way ANOVA, followed by Dunnett’s multiple comparison test, was performed to assess differences between KO and WT cells. (**d**) LECs were allowed to form a monolayer for 24 h and were stimulated with LPS for 4 h before THP-1 WT, KI, and KO cells (stained with Hoechst 33342) were allowed to adhere to the endothelial monolayer under shaking conditions for 15 min. Bar graphs show mean values ± standard deviation of relative fluorescence units (n = 3). Two-way ANOVA, followed by Dunnett’s multiple comparison test, was performed to assess differences between KO, WT, and KI cells (** *p* < 0.01; *** *p* < 0.001; **** *p* < 0.0001). (**e**) Primary human umbilical vein endothelial cell (HUVEC) monolayers were pretreated with conditioned media from LPS-stimulated whole blood for 4 h, and relative intercellular adhesion molecule-1 (ICAM-1) mRNA expression was assessed using the comparative CT method (2^−ΔΔCT^). Bar graphs show mean values ± standard deviation (n = 4). A *t*-test was performed to assess differences between untreated and treated cells. (**f**) After endothelial cell activation for 4 h under flow (5 dyn/cm^2^), cell adhesion of KO, WT, and KI cells was visualized by brightfield microscopy for 15 min with one image taken every 15 s. The scale bar represents 100 µm.

**Figure 2 cells-12-01425-f002:**
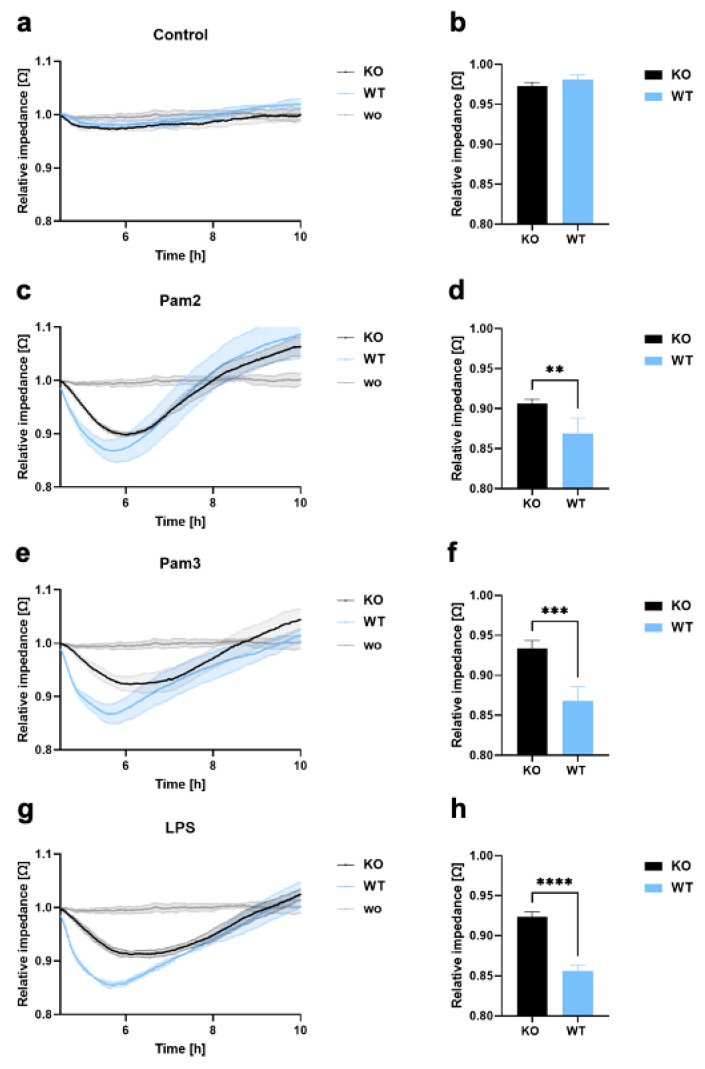
Endothelial barrier disruption and transmigration in THP-1 WT and KO cells. (**a**–**h**) LECs were seeded onto ECIS arrays (96W20idf PET) to form an endothelial monolayer, and 1 × 10^5^ THP-1 KO or WT cells were added (**a**,**b**), either without a ligand or with (**c**,**d**) Pam2, (**e**,**f**) Pam3, or (**g**,**h**) LPS. Monolayer disruption was documented in real time by impedance measurements using the ECIS system (9600Z). Time course diagram and bar graphs at 6 h after treatment showing mean values ± standard deviation (n = 4). A *t*-test was performed to assess differences between KO and WT cells (** *p* < 0.01; *** *p* < 0.001; **** *p* < 0.0001).

**Figure 3 cells-12-01425-f003:**
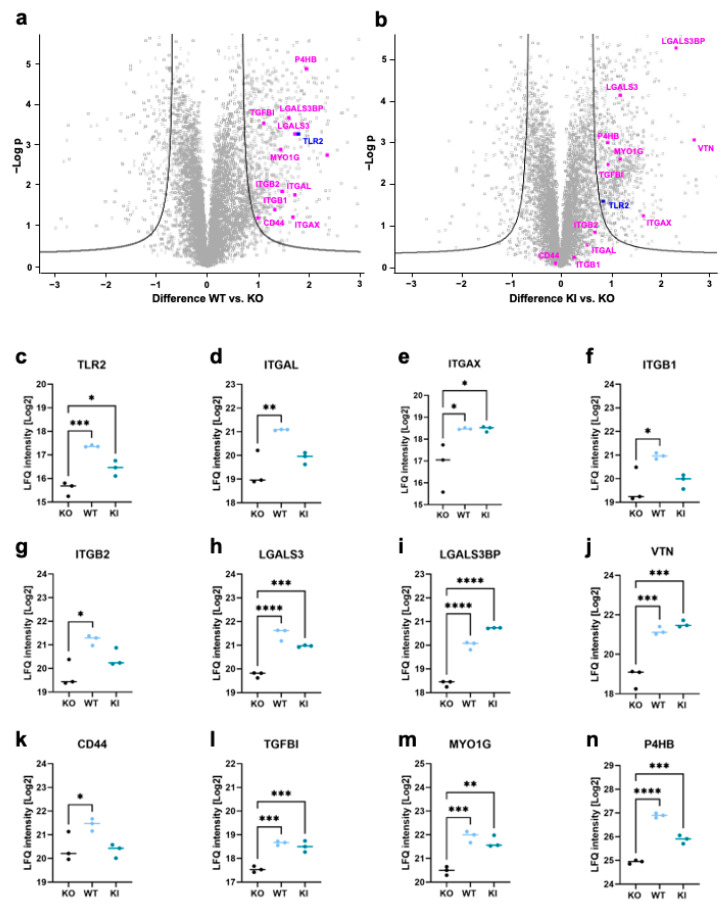
Proteomics analysis of THP-1 WT, KO, and KI cells. (**a**,**b**) Volcano plots showing significant regulations (FDR = 0.05, S0 = 1) of more than 200 proteins in the whole cell lysates from WT and KI compared to KO cells. The difference in LFQ protein abundance between WT/KI and KO (*x*-axis) was plotted against its -log p-value (*y*-axis) (n = 3). (**c**–**n**) Protein signature characteristic for THP-1 WT, KO, and KI cells of (**c**) TLR2, (**d**) Integrin αL, (**e**) Integrin αX, (**f**) Integrin β1, (**g**) Integrin β2, (**h**) Galectin-3 (LGALS3), (**i**) Galectin-3-binding protein (LGALS3BP), (**j**) Vitronectin (VTN), (**k**) CD44, (**l**) Transforming growth factor-beta-induced protein ig-h3 (TGFBI), (**m**) Myosin 1G (MYO1G), and (**n**) Protein disulfide-isomerase beta-subunit of prolyl 4-hydroxylase (P4HB) (n = 3). Interleaved scatter plots show multi-parameter corrected significant protein regulations (FDR = 0.05, S0 = 1). Statistical significances and p-values correspond to the statistical tests underlying the volcano plots and are listed in Appendix A (* *p* < 0.05; ** *p* < 0.01; *** *p* < 0.001; **** *p* < 0.0001).

**Figure 4 cells-12-01425-f004:**
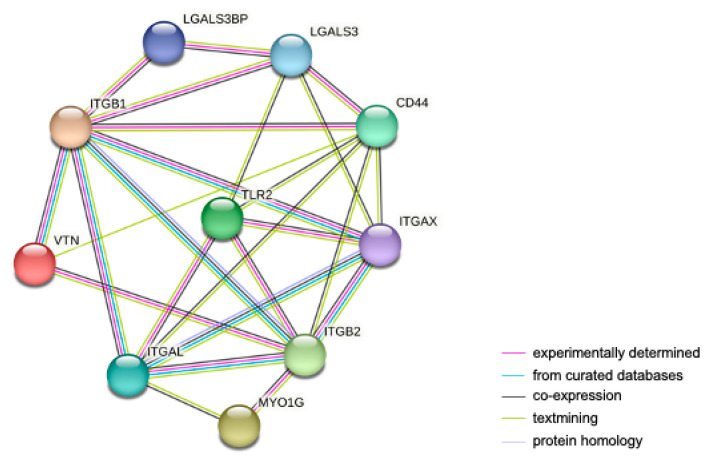
STRING protein–protein interaction network of upregulated genes in THP-1 WT and/or KI vs. KO cells involved in cell adhesion and migration. The network contains 10 nodes and 20 edges. Nodes represent proteins; lines represent protein interactions. The color of the lines represents the evidence of interaction between the proteins.

**Figure 5 cells-12-01425-f005:**
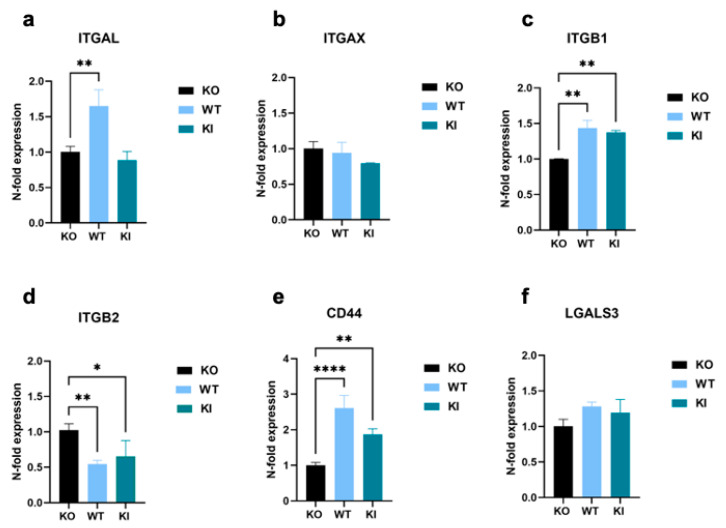
Genomic expression analysis of cell-adhesion-promoting molecules in THP-1 WT, KI, and KO cells. (**a**–**e**) Real-time qPCR showing relative expression levels of (**a**) ITGAL, (**b**) ITGAX, (**c**) ITGB1, (**d**) ITGB2, (**e**) CD44, and (**f**) LGALS3 calculated according to the comparative CT method (2^−ΔΔCT^). Bar graphs show mean values ± standard deviation (n = at least 4). One-way ANOVA, followed by Dunnett’s multiple comparison test, was performed to assess differences between KO, WT, and KI cells (* *p* < 0.05; ** *p* < 0.01; **** *p* < 0.0001).

**Figure 6 cells-12-01425-f006:**
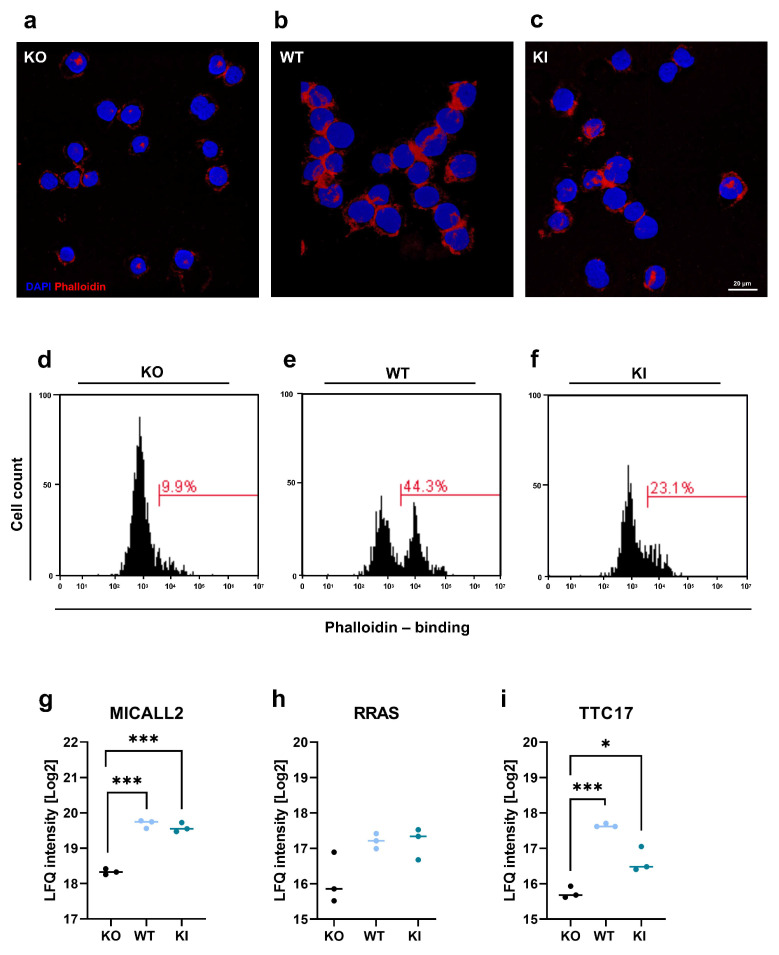
Actin organization in THP-1 WT, KI, and KO cells. (**a**–**c**) After harvesting THP-1 WT, KI, and KO cells, they were stained with phalloidin conjugated to AlexaFluor 635 to visualize F-actin (red), and cell nuclei were stained with DAPI (blue). The scale bar represents 20 µm. (**d**–**f**) Cells were harvested and stained with phalloidin and analyzed by flow cytometry. Histograms show phalloidin-binding in THP-1 KO, WT, and KI cells. (**g**–**i**) Protein signature characteristic for THP-1 WT, KO, and KI cells (n = 3) of (**g**) MICAL-like protein 2 (MICALL2), (**h**) Ras-related protein R-Ras (RRAS), and (**i**) Tetratricopeptide repeat protein 17 (TTC17). Interleaved scatter plots show multi-parameter corrected significant protein regulations (FDR = 0.05, S0 = 1). Statistical significances and *p*-values correspond to the statistical tests underlying the volcano plots and are listed in Appendix A (* *p* < 0.05; *** *p* < 0.001).

## Data Availability

Data presented in this study are available on request from the corresponding authors.

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
