# Peer review of "Beyond Pattern Recognition: TLR2 Promotes Chemotaxis, Cell Adhesion, and Migration in THP-1 Cells"

_cells, 2023, doi:10.3390/cells12101425_

Round 1

Reviewer 1 Report

The paper is well written: the Introduction is comprehensive, the Results described clearly, Methods are accurately described and all the information regarding the models and the procedures are detailed. In particular the test to evaluate the destruction of the endothelial barrier is elegant and informative even if it is very simple. To be considered for publication, some minor changes are required.

1.      At the end of the Introduction, the Authors could explain more clearly the interest in the role of the TLR2 per se, as they then clarify in the Results

2.      lines 284-288 can be considered omitted because they are information already present in the Introduction and Methods

3.      LPS presumably remained in the supernatant used for the chemotaxis test. It should be described and discussed whether the cellular model of THP1 express TLR4, which could activate a cascade in response to binding with residual LPS in the supernatant used for stimulation. Even if the experiments are based on the comparison of models expressing or not expressing the TLR2, since the Authors claim that unstimulated TLR2 per se plays a role in this homing process based on several in vitro test conducted in such condition, they should add a brief supplementary explanation in the Results and a full explanation in the Discussion.

4.      Please, clarify the choice of using Pam2 and Pam3 for the stimulation of endothelial cells

5.      In graphs 1c-e it is better to specify that it is a supernatant of cells stimulated with LPS and not of LPS

6.      It would be interesting to add in the text the relevant data e/o numbers relating to the uncommon proteins found between WT and KI and how many uncommon between WT, KI vs KO, to clarify how much the KI model mirrors the condition of the starting cell

7.      A short concluding sentence relating to the Results of section 3.2 would be helpful

8.      It is not clear why the assay of actin polymerization in the three cell models was conducted in the absence of stimulation

9.      The Discussion deepens and analyzes exhaustively all the results obtained. However, it would be useful in the concluding paragraph to attempt to add some specific summary conclusions relating to protein interactions and the observed phenomena

1.   the Authors could speculate in the Discussion the lack of correlation between the protein and mRNA levels found differentially expressed in the three models

Author Response

Point 1: At the end of the Introduction, the Authors could explain more clearly the interest in the role of the TLR2 per se, as they then clarify in the Results.

Response 1: Thanks for this comment. As suggested, we have added one more sentence at the end of the introduction describing the interest of TLR2 per se: We found that TLR2 per se, without prior activation, plays a significant role in the homing process including chemoattraction, cell adhesion, and cell migration. 

Point 2: Lines 284-288 can be considered omitted because they are information already present in the Introduction and Methods.

Response 2: Thank you for your suggestion. We have omitted the part and start with: To investigate whether TLR2 plays an important part in rolling, adhesion, and cell migration/transmigration during inflammatory events, we used a monocytic THP-1 WT, a KO, and a stable KI cell line.

Point 3:  LPS presumably remained in the supernatant used for the chemotaxis test. It should be described and discussed whether the cellular model of THP1 express TLR4, which could activate a cascade in response to binding with residual LPS in the supernatant used for stimulation. Even if the experiments are based on the comparison of models expressing or not expressing the TLR2, since the Authors claim that unstimulated TLR2 per se plays a role in this homing process based on several in vitro test conducted in such condition, they should add a brief supplementary explanation in the Results and a full explanation in the Discussion.

Response 3: Thank you for this valid point. We have added the following to our discussion: In addition to LPS-induced cytokine and chemokine production, residual LPS itself in the endothelial cell supernatant can activate pattern recognition receptors such as TLR4, TLR2 and the nucleotide binding oligomerization domain containing 1 (NOD1) receptor expressed on THP-1 cells [47,48]. To this end, MyD88-mediated LPS/TLR4 crosstalk with TLR2 has also been described to enhance and stabilize NF-κB and ICAM-1 expression [49]. However, unlike primary monocytes, THP-1 cells express very low levels of CD14, and consequently THP-1 cells hardly respond to LPS via the TLR4 pathway [50]. Furthermore, we assume that the expression levels of the aforementioned receptors are the same in our THP-1 cell lines, except for TLR2, since no differences were detected in our proteomic analysis. This supports the theory that TLR2 is an essential chemoattractant sensor and consequently contributes to cell adhesion and migration [51].

Point 4: Please, clarify the choice of using Pam2 and Pam3 for the stimulation of endothelial cells.

Response 4: As mentioned in the methods, Pam2 and Pam3 are typical TLR2 ligands. Since TLR2 is also expressed on endothelial cells, we wanted to see if there was a difference when activating the endothelium with different TLR ligands. We added the explanation to the results: Cells were left untreated or treated with LPS, which is a potent stimulator of TLR4, and Pam2 and Pam3, which are TLR2 ligands, and THP-1 KO and WT cells were added directly to the endothelial cells.

Point 5: In graphs 1c-e it is better to specify that it is a supernatant of cells stimulated with LPS and not of LPS.

Response 5: This is a good input; we clarified the difference in figure 1b and c by adding Sup. Wo and Sup. LPS for the chemoattraction/migration assay graphs.  

Point 6: It would be interesting to add in the text the relevant data e/o numbers relating to the uncommon proteins found between WT and KI and how many uncommon between WT, KI vs KO, to clarify how much the KI model mirrors the condition of the starting cell.

Response 6: We have changed the sentences and added a more detailed description for clarification: In general, approximately 7200 proteins were identified in our samples, of which 858 proteins were significantly regulated when comparing WT vs. KO and 951 proteins were significantly regulated when comparing KI vs. KO (FDR=0.05, s0=1). Furthermore, 236 common proteins were found to be significantly upregulated in WT and KI vs. KO (Table S1).

Point 7: A short concluding sentence relating to the Results of section 3.2 would be helpful.

Response 7: Thanks for this remark, we added a concluding sentence: In summary, we found that certain integrins are dependent on TLR2 expression and also identified novel proteins such as MYO1G or P4HB that are influenced by TLR2.

Point 8: It is not clear why the assay of actin polymerization in the three cell models was conducted in the absence of stimulation.

Response 8: Here we wanted to show that TLR2 has an influence without prior activation as we also showed the different protein expression of the three cell lines without activation and identified actin regulatory proteins. Furthermore, LPS would influence actin polymerization, but this was beyond our scope.

Point 9: The Discussion deepens and analyzes exhaustively all the results obtained. However, it would be useful in the concluding paragraph to attempt to add some specific summary conclusions relating to protein interactions and the observed phenomena.

Response 9: We have revised the discussion and added a more specific conclusion concerning the protein interactions: Overall, by comparing these three monocytic cell lines, we were able to demonstrate that TLR2 is a key mediator of cell adhesion and cell migration to the endothelium, even without prior stimulation. We discovered altered mRNA and protein expression patterns of several adhesion molecules such as integrins, including ITGB1 and ITGAL, CD44, and LGALS3 dependent on TLR2 expression. In addition, novel TLR2-dependent proteins such as P4HB and MYO1G, which are essential for the homing process, were uncovered and new connections between TLR2 and actin polymerization were identified. Our results suggest that TLR2 in monocytes may be an important contributor not only in the initiation of inflammation, but also in adhesion and cell migration by influencing the protein expression of integrins, adhesion molecules and actin regulatory proteins.

Point 10: The Authors could speculate in the Discussion the lack of correlation between the protein and mRNA levels found differentially expressed in the three models.

Response 10: Thank you for the input. We have discussed the lack of correlation in more detail (+additional references).

For the integrins: In contrast to the proteomics data, the mRNA level was significantly decreased in cells expressing TLR2. This led us to hypothesize that the heterogeneity may be due to post-translational modifications and/or alternative splicing, which is not an uncommon event to observe [41,43]. More than 400 types of modifications have been identified in proteins involved in adhesion and migration, including integrins [58]. Overall, these results show that TLR2, even in the absence of ligand activation, affects the protein ex-pression of certain integrins.

For CD44:. A recent study revealed that TLR2 engagement alters CD44 expression and modulates alternative splicing of CD44 pre-mRNA, which may explain the poor correlation between mRNA and protein levels [65].

Reviewer 2 Report

What do you mean with the largest group of leukocyte? Is this correct for blood monocytes?

 Line 38: replace leukocytes with monocytes as you are talking about monocyte migration.

 Line 357-358: Interestingly, THP-1 WT significantly enhanced endothelial 357 monolayer disruption compared to KO cells”. What about THP-1 KI cells?

 Line 532: “We found that CD44 was significantly elevated 532 in WT vs. KO, but not at all in KI vs. KO cells” how could you explain this?

 The discussion part needs revision with focus on the interpretation of findings rather than repeating the results.

 Would you expect similar or different results if using primary monocytes instead of the THP-1

Author Response

Point 1: What do you mean with the largest group of leukocyte? Is this correct for blood monocytes?

Response 1: Thank you for your input, we have clarified the sentence: Monocytes, which comprise approximately 10% of total human leukocytes, play a fundamental role in protective immunity and are involved in both the initiation and resolution of inflammation.

Point 2:  Line 38: replace leukocytes with monocytes as you are talking about monocyte migration.

Response 2: We have replaced leukocytes with monocytes.

Point 3: Line 357-358: “Interestingly, THP-1 WT significantly enhanced endothelial 357 monolayer disruption compared to KO cells”. What about THP-1 KI cells?

Response 3: Thank you for mentioning this; we have now added a supplementary figure where THP-1 KI is included and added the following sentence to the results: Supplementary Figure S2 demonstrates that the endothelial barrier disruption with THP-1 KI cells is not as intense as with WT cells, but stronger than with KO cells, which is explained by the lower TLR2 expression.

Point 4: Line 532: “We found that CD44 was significantly elevated 532 in WT vs. KO, but not at all in KI vs. KO cells” how could you explain this?

Response 4: Thank you for raising this question. We added following sentences to the discussion: There is growing evidence that CD44 affects TLR2 signaling [63,64]. However, since we see that CD44 is only upregulated in WT THP-1 and not in KI, this may suggest that a certain amount of TLR2 is required to lead to higher CD44 protein expression.

Point 5:  The discussion part needs revision with focus on the interpretation of findings rather than repeating the results.

Response 5: Thank you for this suggestion. We have revised the discussion focusing on a more detailed interpretation and included additional references (see track changes).

Point 6: Would you expect similar or different results if using primary monocytes instead of the THP-1?

Response 6: Thank you, this is a valid question. There are mouse models with TLR2 KO (McCoy et al., 2021) and human TLR2 knockdown (Chang et al., 2007), but a complete KO is very difficult to facilitate with primary monocytes due to their short lifespan of about 5 days (Patel et al., 2017). THP-1 cells are a common model to assess the modulation of monocyte and macrophage activities. However, we can only speculate whether we would see a similar trend with primary monocytes.